# Human Red Blood Cells as Oxygen Carriers to Improve Ex-Situ Liver Perfusion in a Rat Model

**DOI:** 10.3390/jcm8111918

**Published:** 2019-11-08

**Authors:** Daniele Dondossola, Alessandro Santini, Caterina Lonati, Alberto Zanella, Riccardo Merighi, Luigi Vivona, Michele Battistin, Alessandro Galli, Osvaldo Biancolilli, Marco Maggioni, Stefania Villa, Stefano Gatti

**Affiliations:** 1General and Liver Transplant Surgery Unit, Fondazione IRCCS Ca’ Granda, Ospedale Maggiore Policlinico, 20019 Milan, Italy; 2Department of Pathophysiology and Transplantation, Università degli Studi of Milan, 20019 Milan, Italy; alberto.zanella1@unimi.it; 3Department of Anesthesia and Critical Care, Fondazione IRCCS Ca’ Granda, Ospedale Maggiore Policlinico, 20019 Milan, Italy; alesantini85@gmail.com (A.S.); caterina.lonati@gmail.com (C.L.); preclinica@policlinico.mi.it (L.V.); battistin.michele@gmail.com (M.B.); alexgalli@hotmail.com (A.G.); osvaldo.biancolilli85@gmail.com (O.B.); 4Center for Preclinical Research, Fondazione IRCCS Ca’ Granda, Ospedale Maggiore Policlinico, 20019 Milan, Italy; riccardo.merighi92@gmail.com (R.M.); stefano.gatti@gmail.com (S.G.); 5Pathology Department, Fondazione IRCCS Ca’ Granda, Ospedale Maggiore Policlinico, 20019 Milan, Italy; marco.maggioni@policlinico.mi.it; 6Department of Transfusion Medicine and Hematology, Fondazione IRCCS Ca’ Granda, Ospedale Maggiore Policlinico, 20019 Milan, Italy; stefania.villa@policlinico.mi.it

**Keywords:** normothermic machine perfusion, rat, human red blood cells, oxygen consumption, oxygen delivery

## Abstract

Ex-situ machine perfusion (MP) has been increasingly used to enhance liver quality in different settings. Small animal models can help to implement this procedure. As most normothermic MP (NMP) models employ sub-physiological levels of oxygen delivery (DO_2_), the aim of this study was to investigate the effectiveness and safety of different DO_2_, using human red blood cells (RBCs) as oxygen carriers on metabolic recovery in a rat model of NMP. Four experimental groups (n = 5 each) consisted of (1) native (untreated/control), (2) liver static cold storage (SCS) 30 min without NMP, (3) SCS followed by 120 min of NMP with Dulbecco-Modified-Eagle-Medium as perfusate (DMEM), and (4) similar to group 3, but perfusion fluid was added with human RBCs (hematocrit 15%) (BLOOD). Compared to DMEM, the BLOOD group showed increased liver DO_2_ (*p* = 0.008) and oxygen consumption ($\dot{VO}$_2_) (*p* < 0.001); lactate clearance (*p* < 0.001), potassium (*p* < 0.001), and glucose (*p* = 0.029) uptake were enhanced. ATP levels were likewise higher in BLOOD relative to DMEM (*p* = 0.031). $\dot{VO}$_2_ and DO_2_ were highly correlated (*p* < 0.001). Consistently, the main metabolic parameters were directly correlated with DO_2_ and $\dot{VO}$_2_. No human RBC related damage was detected. In conclusion, an optimized DO_2_ significantly reduces hypoxic damage-related effects occurring during NMP. Human RBCs can be safely used as oxygen carriers.

## 1. Introduction

Liver machine perfusion (MP) was introduced in the clinical setting by Guarrera and colleagues [1] in 2009. Based on its ability to recondition, evaluate, and preserve liver grafts, MP showed a particular potential in reverting the detrimental impact of extended criteria donors (ECD) on post-liver transplant (LT) outcome and quality of life [2,3]. Among MPs, normothermic machine perfusion (NMP) represents the most promising technology due to its higher evaluation potential [4]. Further, NMP can be used in preclinical settings to evaluate other preservation or treatment techniques outside LT. In this last setting, NMP is defined as normothermic machine reperfusion (NMRP).

Since reactions elicited during ex-situ dynamic perfusion are largely unknown, technical and biological aspects of liver MP need to be extensively investigated through committed research. A number of animal models have been developed for this purpose. Because of their immediate translational value (e.g., appropriate human-like liver size), swine models have been broadly used in the start-up phase to rapidly translate preclinical results into the clinical setting. Conversely, small animal models can better be exploited to investigate subcellular mechanisms and changes associated with MP. To this purpose, based on low cost, reproducibility, and better understanding of subcellular events, rat models have been widely used [5].

Over the last ten years, 94 research papers were based on liver MP use in rodents. We reviewed all those reporting rat ex-situ perfusions with recirculating fluid and 39 of them described NMP/NMRP (Appendix A: a brief review of the literature is provided in Appendix A). A large variability in NMP protocols among different research groups was evident. In particular, the potential usefulness of an oxygen carrier (OxC) during normothermic perfusion was not fully explored. While only few researchers used OxC perfusate for NMP or NMRP [6,7,8,9,10,11,12,13,14,15,16], most of them adopted a non-OxC model. The infrequent use of OxC could be related to the high number of rats needed to be used as blood donors (at least 3-4 animals/experiment) and unavailability of other oxygen carriers. However, while adequate oxygen delivery (DO_2_) was observed in the absence of an oxygen carrier, the increased oxygen consumption ($\dot{VO}$_2_) needed to control the reperfusion injury suggests the opportunity to increase DO_2_ during reperfusion [8,17]. To this purpose, two main strategies may be adopted: use of non-cellular hemoglobin [18] or employment of other sources of blood [13]. While the first is limited by the restricted availability of these products, human blood cells can be easily procured for experimental use.

The aim of present research was optimization of procurement and perfusion procedures to obtain a safe and reproducible rat model of NMP. The study examined the potential protective role of improved DO_2_ during NMP and evaluated the safety and efficacy of human red blood cell use. Indeed, the use of non-murine red blood cells can allow for the reduction of the number of animals/experiments in full respect of the 3Rs principles (Refinement, Reduction, and Replacement) [19].

## 2. Material and Methods

### 2.1. Animals and Study Design

Experiments were performed at the Center for Surgical Research, Fondazione IRCCS Ca’ Granda Ospedale Maggiore Policlinico of Milan. Italian Institute of Health approved the experimental protocol (number 568EB.1). All animals received humane care in compliance with the Principles of Laboratory Animal Care formulated by the Federation of European Laboratory Animal Science Associations [20].

Adult Sprague-Dawley male rats weighing 240 to 330 g (Envigo RMS. S.R.L, Udine, Italy) were housed in a ventilated cage system (Tecniplast S.p.A., Varese, Italy) at 22 ± 1 °C, 55 ± 5% humidity, on a 12 h dark/light cycle, and were allowed free access to rat chow feed and water ad libitum. All efforts were made to minimize suffering.

A schematic workflow diagram of the investigation is shown in Figure 1. Twenty rats were randomly assigned to four experimental groups (*n* = 5/group): (1) *DMEM* whose livers were subjected to in-situ cold flushing, procured, cold stored for 30 min, and then ex-situ perfused for 120 min with a perfusion fluid without an oxygen carrier; (2) *BLOOD*, whose livers were subjected to all the procedures of the DMEM group, but NMP perfusion fluid included human red blood cells (RBC) as an oxygen carrier; (3) *Cold storage*, whose livers were subjected to in-situ cold flushing, procured, and then subjected to 30 min of cold storage; (4) *Native*, whose livers were procured immediately after anesthesia and suppressed.

### 2.2. Blood Group Typing and Compatibility Testing

Human blood used was collected in CPD (citrate-phosphate- dextrose), resuspended in SAGM (saline-adenine-glucose-mannitol), and leukoreduced at final hemtocrit of 60 ± 5%.

Blood samples of the Native group were tested for blood group typing. Blood was collected in K3-EDTA test tube and tested with 0 Rh (D) negative and AB Rh (D) negative human packed leukoreduced red blood cells and plasma. Direct blood group typing was performed with manual method using ABO Ortho BioVue Card (Ortho Clinical Diagnostics, Pencoed, UK) on 3% blood cells diluted in saline solution. Indirect blood group typing was manually performed on fresh plasma through Reverse Diluent Ortho BioVue Card with standard blood cells A1, B, and 0 (Affirmagen Ortho). We repeated the same analyses at 4 °C to enhance ABO antibody mediated activity.

Major compatibility (recipient serum versus donor red blood cells) and minor compatibility (donor plasma with the recipient red cells) were manually performed using Ortho BioVue with anti-IgG, C3d polyspecific Cards.

All rats resulted 0 Rh (D) negative at direct typing. Indirect blood group typing showed in one case anti-A and anti-B antibodies, while in two cases only anti-A antibodies were detected. Major compatibility was negative with 0 Rh (D) negative human red blood cells, while positive with AB Rh (D) negative donors. Minor compatibility was positive with both 0 Rh (D) negative and AB Rh (D) negative plasma of human donors. According to these results, in order to have the greatest incompatibility to highlight any damage from hemolysis, we decided to use AB Rh (D) positive packed leukoreduced red blood cells.

### 2.3. Reagents and Instruments

All experiments were carried out in sterile conditions and all instruments/drugs were kept sterile until use. The drugs, reagents, and instrumentation necessary to conduct this protocol are shown in Appendix A.

### 2.4. Anesthesia, Surgery, and In-Situ Perfusion

A schematic overview of all experimental procedures is provided in Appendix A. Rats were anesthetized by intraperitoneal injection of 80 mg/kg of sodic thiopentale and maintained in spontaneous breathing with an O_2_ enriched air mixture by an open mask. The surgical procedure was carried on with a double-headed surgical microscope OPMI 1-F (Zeiss West Germany, Oberkochen, Germany). Procedure began with a xifo-pubic and bilateral subcostal incision. The hepatic pedicle was then exposed, and the bile duct was dissected. It was distally ligated with a 8-0 nylon tie and a customized cannula (tip: 24 G cannula Braun, Bethlehem, PA, USA; tube: PE-50, BD-Clay Adams, Becton, UK) was inserted through a choledotomy; the cannula was secured with a proximal 8-0 nylon tie. Subsequently, the portal vein was cannulated. Then, the pyloric vein was ligated (8-0 nylon) and transacted, and the main trunk of the portal vein was dissected until distally to the splenic vein. The portal vein was encircled with three 5-0 silk tie and unfractioned heparin (2 IU/g diluted in 1 mL of normal saline) was administered by tail IV injection. After three minutes, we ligated the portal vein distally to the splenic vein insertion and a 16 G cannula was inserted by venipuncture until the tip of the cannula reached the portal bifurcation. A blood retrograde flush was performed, and the in-situ perfusion system was connected. Then, the cannula was secured with a two silk tie and gently retracted to avoid obstruction of the right lobe portal vein branch. Rapid sternotomy was performed, the heart was removed, and the inferior vena cava (IVC) was transected. In-situ cold perfusion (4 °C) was started with 35 mL of Celsior solution (IGL, Lissieu, France) at a pressure of 30 cmH_2_O. At the end of the perfusion, hepatectomy was completed and the liver graft was stored in 4 °C Celsior solution for 30 min. At the end of the cold storage and before NMP connection, the backtable was completed to free the superior vena cava (the liver was maintained on a refrigerated surface). All surgical times were recorded.

### 2.5. Surgical Procedure and In-Situ Perfusion

All procedures were carried out by an expert surgeon trained in microsurgery and ex-situ perfusion models. Time periods required for bile duct cannulation (from laparotomy to cannula placement in the bile duct), portal cannulation (from bile duct cannulation to portal cannula placement), in-situ cold-flush, liver procurement (from flushing to procurement), warm ischemia time (from the end of in-situ flush to the start of the cold storage), and liver connection to NMP (from end of SCS to connection to the circuit) were examined as indexes of technical success and reproducibility of the surgical procedure. The rats that were randomized to native group (*n* = 5) were not included in this analysis because they did not undergo any surgical procedure. There were no statistically significant differences among the “surgical” groups in terms of surgical preparation 37 ± 1 min for DMEM, 35 ± 2 min for BLOOD, and 35 ± 2 min for SCS. Cold flush lasted 60 ± 4 sec for DMEM, 71 ± 18 sec for BLOOD, and 61 ± 3 sec for SCS, while warm ischemia time 16 ± 1 min for DMEM, 16 ± 3 min for BLOOD, and 16 ± 2 min for SCS.

### 2.6. Ex-Situ Liver Perfusion

#### 2.6.1. Perfusion Fluids Preparation

Perfusate composition is shown in Appendix A. DMEM perfusate was obtained by adding to DMEM, 4% human albumin, streptomycin and penicillin, glutamine, and insulin for a total volume of perfusate of 100 mL. Perfusion fluid was obtained according to Op den Dries and colleagues [13]. We added to DMEM 4% human albumin, streptomycin and penicillin, glutamine, insulin, heparin, and 15% hematocrit human RBC for a global volume of perfusate of 100 mL.

Human red cells were provided by the Blood Bank of our Institution (Department of Transfusion Medicine and Hematology). A packed red blood cell concentrate was prepared by centrifugation from one unit of a whole blood donation, which was unsuitable for human use because of an insufficient volume collected. The same unit (blood group: AB Rh+) was used throughout the experiment.

#### 2.6.2. Normothermic Machine Perfusion Setup

The perfusion system (Figure 2) consisted of a customized circuit derived from the isolated lung perfusion system (Hugo Sachs Elektronik, Harvard Apparatus, March-Hugstetten, Germany) implemented and described by our laboratory [21,22,23,24]. It consisted of a reusable heated glass reservoir, a heated bubble trap, a circulating tube derived by an infusion system, an octagonal peristaltic pump, a polystyrene lid, and a single-use artificial lung (Appendix A). The liver was placed onto the glass chamber modified to let the liver laid on the diaphragmatic surface on a modeled ad hoc, perforated parafilm. The liver was connected to the circuit through the portal vein cannula and the bile duct cannula to a 2 mL tube to collect bile. The chamber was closed to maintain humidity. Temperature inside the chamber was recorded with a 1.3 mm probe. The heat-exchanger was set at 40 °C to obtain a graft temperature of 37 °C. We carefully avoided air embolism during priming and liver connection. The artificial lung was ventilated with a 200 mL/min flow, with 95% FiO2 and 5% FiCO_2_.

Circuit hemodynamic parameters were monitored using the Colligo system (Elekton, Milan, Italy). Portal vein and pre-lung pressures were recorded every 5 min during rewarming, and every 30 min during normothermic machine perfusion. Vascular resistances were calculated as mean portal vein pressure divided by blood flow in the portal vein (mL/min). Temperature was monitored by a thermo probe located between the lobes of the liver.

#### 2.6.3. Ex-Vivo Liver Perfusion Protocol

The NMP-protocol lasted 150 min and divided into two phases. The first 30 min was called “rewarming”, followed by 120 min of normothermic perfusion (Figure 3). In brief, after 30 min of static cold storage, the graft was connected to the NMP and the perfusion began. The portal flow was set at 5 mL/min and was increased by 5 mL/min every 5 min up to 30 mL/min or portal pressure of 8 cmH_2_O. At the beginning of the rewarming phase, the heat exchanger was set at 30 °C and the temperature was raised up to 40 °C to obtain a liver temperature of 37 °C after 30 min. During the normothermic phase temperature was maintained at 37 °C and portal flow remained unchanged.

### 2.7. Perfusate Analysis

Perfusate samples were collected soon after grafting ex-situ reperfusion and then every 30 min during the normothermic phase. The concentration of gas, metabolites, hyaluronan (HA), electrolytes, hemoglobin, and cells were evaluated. Pre-liver stopcock was used for perfusate sampling, while post liver perfusate was collected directly from IVC. Bile was collected by gravity directly into a test tube and weighted.

### 2.8. Sample Processing and Analysis

Perfusate samples were centrifugated at 1500 rpm for 15 min at 4 °C (Haereus Multifuge 33R, Thermo Fisher Scientific, Whaltham ). Supernatants were collected and stored at −20 °C for mediator concentration evaluation, whereas cell pellets were suspended in erythrocyte lysis buffer (0.155 M NH4Cl, 10 mM KHCO3, 0.1 mM Na2EDTA, pH 7.4; all from MilliporeSigma, Frankfurter, Germany) and incubated for 10 min at 4 °C. A 10-min centrifugation at 2000 rpm was then performed. Recovered supernatants were used for free hemoglobin assessment, whereas pellets were suspended in 0.25 mL 13 PBS solution (MilliporeSigma, Frankfurter, Germany) for cell count and characterization.

Using a gas analyzer, acid-base balance (pH, pCO_2_, pO_2_), electrolytes (K^+^, Na^+^, Cl^−^, Ca^2+^), and metabolites (glucose and lactate) were evaluated (ABL 800 Flex; A. De Mori Strumenti, Milan, Italy).

At the beginning and end of the normothermic phase, perfusate samples were investigated for hepatocyte integrity (alanine-aminotransferase, ALT; aspartate-aminotransferase, AST; Lactate dehydrogenase, LDH), cholangiocellular damage (ALP, GGT), and blood count.

The concentration of total HA in outflow perfusate was determined by ELISA using a commercially available kit (sensitivity, 0.068 ng/mL; low standard, 0.625 ng/mL; R&D Systems, Minneapolis, MN, USA). The procedure was validated in preliminary experiments by using HA preparations with a known concentration and known average molecular mass (Select-HA HiLadder and LoLadder; Hyalose, Oklahoma City, OK, USA) [21,22,23,24].

Sodium Citrate concentration was measured by enzymatic assay using spectrophotometry (Citrate Assay Kit, Sigma-Aldrich, Saint Louis, USA). Millipore Amicon Ultra 0.5 mL 20 KDa were used to remove proteins. Thereafter, deproteinized samples were diluted 1:2 with citrate assay buffer and centrifugated at 14,000× *g*.

### 2.9. Tissue Analysis

Tissue samples were taken at the end of NMP in DMEM and BLOOD groups, at the end of cold storage in SCS group, and soon after laparotomy in group Native. Tissue samples (*n* = 8) were collected from the right median lobe: 1 sample was used for wet-to-dry ratio (W/D), 1 formalin fixed. Liver grafts were weighted at the end of static cold storage and the end of NMP.

### 2.10. Wet-to-Dry Ratio

A biopsy was weighed with an analytical balance, dried in an oven at 50 °C for 48 h, and then weighted [25]. Wet-to-dry ratio (W/D) was calculated and used as an index of edema. Livers procured from native and cold ischemia group were used as controls.

### 2.11. ATP Content Assessment

Liver samples were homogenized in trichloroacetic acid (MilliporeSigma, Frankfurter, Germany), then subjected to 10 min of centrifugation at maximum speed at 4 °C. Supernatants were diluted 1:30 using 0.1 M Tris-acetate, pH 7.75 (MilliporeSigma, Frankfurter, Germany). Next, 10 mL of each sample were dispensed in a blank 96-well plate and 90 mL of luciferin/luciferase reagent were added (Enliten ATP Assay System; Promega, Madison, WI, USA). Bioluminescent signals were immediately detected with a luminometer (Glomax Luminometer; Promega, Madison, WI, USA). ATP concentration was calculated by using a standard curve that ranged from 10,211 to 1025 M (rATP 10 mM; Promega, Madison, WI, USA). Results were expressed as concentration/wet tissue weight.

### 2.12. Histology

Liver samples were fixed in 4% formalin. Formalin-fixed-paraffin-embedded samples were stained with hematoxylin-eosin, Masson’s trichrome, Periodic Acid Schiff (PAS) and reticulin histochemical staining, and CD31 immunohistochemical staining. Thirty random fields per slide were investigated to determine the necrosis area.

To evaluate liver tissue integrity, the histological samples were scored according to Brockman and colleagues [26] who demonstrated concordance among NMP results, histopathological analyses, and liver viability.

### 2.13. Evaluation of Liver Metabolism

The above-mentioned parameters and trends (glucose, lactates, potassium, and sodium citrate) were included in the liver metabolism evaluation. Citrate is metabolized by hepatocyte in the perivenular zone of hepatic lobule and its eventual decrease during the NMP procedure reflects active metabolism [27]. Equally, potassium uptake after the ischemic phase is indicative of an adequate reactivation of Na-K ATP dependent pumps in viable hepatocytes. The uptake ratio ((C_start_−C_end_)/C_start_) was used to describe the variation of metabolic parameters during NMP.

Oxygen consumption ($\dot{VO}$_2_) was measured using the modified Fick equation (Appendix A). Pre-liver perfusate samples were intended as O_2_ enriched perfusate (arterial blood of the Fick equation), whereas post-liver perfusate samples, collected directly from the IVC, were used for the calculation of venous oxygen content in the Fick equation.

### 2.14. Statistical Analysis

Statistical analysis was performed using SPSS statistics 25.0 software (IBM Corporation, Armonk, USA). Results are expressed as mean ± SEM. The sample size was determined considering a statistical test power of 0.80, with a alpha value of 0.05 and a minimum difference among groups of 0.40 with an expected SD of 0.20. Metabolic and viability parameters were analyzed by using 1-way repeated measures ANOVA. Conversely, 1-way ANOVA or Kruskal–Wallis test were used to evaluate ATP and uptake ratio results. Finally, linear regression analysis was used to correlate variables. Values of *p* < 0.05 were considered statistically significant.

## 3. Results

A total of 20 experiments were consecutively performed. Nineteen grafts are included in the analysis because one liver graft from the BLOOD group was discarded due to air embolism.

### 3.1. Perfusate Composition and Hemodynamic During NMP

Liver perfusion was started when the pH of the solution was at least 7.2. Table 1 shows the gas-analyses characteristics of the perfusate before graft connection to the NMP. Besides the differences due to the presence of red blood cells in the BLOOD group, pH, glucose, and bicarbonate concentration were statistically different at baseline gas-analyses between the groups. Baseline citrate concentration was 0.37 ± 0.05 mmol/L in the BLOOD group, while it was not detectable in the DMEM group.

During the rewarming phase, in which flow rate, temperature, and gas flow were gradually increased up to target value, there was a decrease in portal resistance in both groups (Figure 4). While in the rewarming phase portal resistance was not statistically different between the two groups (*p* = 0.105), during the normothermic phase, consistently with higher viscosity of the perfusing solution, it was higher in the BLOOD group (*p* = 0.015).

### 3.2. Markers of Hepatocellular and Cholangiocellular Damage

Hepatocellular integrity was evaluated during the normothermic phase (Table 2). Even if transaminases (AST) and lactate-dehydrogenase (LDH) showed a trend toward increase, this was not significant (*p* = 0.358). Furthermore, there were no differences in AST and LDH levels between the groups (Appendix A). Potassium remained stable in DMEM, while it was reabsorbed in BLOOD (*p* = 0.001). Potassium uptake ratio was also higher in the BLOOD vs. DMEM group (Figure 5A). The total amount of bile collected was higher in BLOOD group (Figure 5B). Hyaluronan decreased during the perfusion (*p* = 0.002) without any differences between the groups (*p* = 0.331).

### 3.3. Evaluation of Liver Tissue Integrity

W/D ratio was not different between DMEM and BLOOD groups (DMEM 3.117 ± 0.136 vs. BLOOD 2.995 ±. 0.85, *p* = 0.208). Compared to SCS group (W/D 2.872 ± 0.128), parenchymal edema was increased in DMEM (vs SCS, *p* = 0.029), while W/D ratio was not statistically different in BLOOD (vs. SCS, *p* = 0.275). Liver histological architecture was preserved in all study groups (Figure 6). Both DMEM and BLOOD showed normal histology and the grade of post-reperfusion injury was not different between the study groups (Figure 5D). As expected, red blood cells were identified within sinusoids and vessels of BLOOD group specimens. Histopathological evaluation showed no sinusoidal obstructions, hemorrhages, or endothelial damage. PAS reaction on tissue samples identified a uniform distribution of glycogen in all hepatic zones of the lobule in BLOOD, while large areas of glycogen consumption were identified in the periportal spaces of DMEM.

### 3.4. Liver Metabolism During NMP

During the normothermic phase, the metabolic function of the liver was evaluated to assess the performance of the two different types of perfusate (Table 2). The DO_2_ of the BLOOD group was higher than in DMEM (DO_2_: DMEM 0.405 ± 0.015 mL/min vs. BLOOD 1.685 ± 0.090 mL/min, *p* < 0.001) (Figure 7A). Equally, $\dot{VO}$_2_ was higher in BLOOD group (*p* = 0.001). While DMEM $\dot{VO}$_2_ did not change through the procedure, the BLOOD $\dot{VO}$_2_ decreased over time (Figure 7B). The energetic load of liver tissue decreased during the 30 min of cold storage (native 1354.2 ± 109.1 pmol/mg vs. SCS 985.9 ± 102.6 pmol/mg, *p* = 0.046). At the end-NMP, ATP levels decreased independently of the presence of OxC (DMEM 327.3 ± 26.6 pmol/mg; BLOOD 565.8 ± 56.6 pmol/mg) compared to both SCS (DMEM vs. SCS, *p* = 0.007 and BLOOD vs. SCS, *p* = 0.012) and native (DMEM vs. native *p* = 0.001 and BLOOD vs. native *p* = 0.003) group, as indicated by a bioluminescent assay of liver homogenates. However, after 120 min of normothermia the grafts perfused with a blood based perfusate showed a statistically significant higher ATP (*p* = 0.031) (Figure 5C). Glucose during NMP was stable in DMEM group, while decreased in BLOOD group (*p* < 0.001), resulting in a greater concentration of glucose in perfusion fluid of DMEM group at the end of the experiment compared to the BLOOD group (*p* = 0.008) (Figure 7C). Glucose uptake ratio was higher in DMEM (*p* = 0.029). Lactates were higher at each time point (*p* = 0.002) and lactates uptake ratio was lower in DMEM (DMEM 0.062 ± 0.359 vs. BLOOD 0.643 ± 0.027, *p* = 0.029) (Figure 7D). Interestingly, during the ischemic phase (DO_2_ = 0 mL/min), the estimated glucose uptake ratio was −0.280 ± 0.116, while the lactates uptake ratio was −0.180 ± 0.89. Citrate uptake ratio was 0.228 ± 0.127 in BLOOD group (it was not detectable in DMEM).

The graft $\dot{VO}$_2_ (*R*^2^ = 0.929; *p* < 0.001) (Figure 8), the end-NMP tissue ATP content (*R*^2^ = 0.622; *p* = 0.020) and the ability of the liver to clear lactates (*R*^2^ = 0.609; *p* = 0.022) and glucose (*R*^2^ = 0.718; *p* = 0.008) were directly related to the amount of oxygen delivered (DO_2_). Equally, an increase in oxygen consumption ($\dot{VO}$_2_) led to an increase in lactates clearance (*R*^2^ = 0.505; *p* = 0.048) and glucose consumption (*R*^2^ = 0.597; *p* = 0.025), while ATP levels showed only a trend toward a significant linear correlation with $\dot{VO}$_2_ (*R*^2^ = 0.454; *p* = 0.067). An increase of 1 pmol/mg of liver ATP produced an increase in lactates uptake ratio of 0.002 ± 0.001 (*R*^2^ = 0.641; *p* = 0.017) and glucose uptake ratio of 0.003 ± 0.001 (*R*^2^ = 0.619; *p* = 0.021).

## 4. Discussion

The present research demonstrates that an optimized DO_2_ reduces liver hypoxic damage during NMP. This significant result was achieved using human RBCs as oxygen carriers. Therefore, in addition to the beneficial effects on metabolic parameters, the study points at the use of human RBCs as a strategy to substantially reduce the number of animals employed in accordance with the 3Rs principles.

NMP is increasingly used in clinical settings [28,29,30] to evaluate, recondition, and preserve liver grafts, and this key strategy could be substantially improved by preclinical research, in particular by rodent models. Further, the use of NMRP, rather than LT to evaluate IRI without the use of a recipient, could reduce the number of animals employed. However, to expand their translational potential, these preclinical models need to be optimized and each experimental step should be carefully implemented [21].

We examined the influence of different perfusion solutions and the use of an oxygen carrier on graft reperfusion and viability. Results show that the addition of human RBCs [12,13] to NMP perfusate increases oxygen delivery and causes a faster and steeper restoration of liver metabolism. Indeed, the increased DO_2_, achieved through the use of human RBCs, led to a more rapid clearance of lactate and partially counteracted the reduction in ATP content at the end of NMP.

Mischinger and colleagues [31] suggested in 1992 that ex-situ normothermic perfused livers do not need high DO_2_. Based on this view, NMP set-up could be simplified by removing the OxC; namely, erythrocytes. Indeed, the general idea was that during NMP there was reduced ex-situ metabolic activity and a mitochondrial inhibition due to hypoxia-induced factors [32,33]. Conversely, other authors suggested that an optimized DO_2_ during perfusion could result in a faster graft recovery after ischemia [34,35]. Autologous rat or human red blood cells can be used as oxygen carriers to increase DO_2_. If rat blood is used, 3 to 4 donors are required to obtain a 15% hematocrit perfusate. Human blood enriched perfusate was used in ex-situ perfusions [12,13] of rat liver grafts, but the authors reported no clear advantage by using an OxC. Further, some papers showed a detrimental impact of human blood used during experiments on different animals [36].

In the present research, the use of human red blood cells and a dedicated membrane lung led to a higher DO_2_ and a subsequent higher $\dot{VO}$_2_ in the BLOOD group. Interestingly, our $\dot{VO}$_2_ data are markedly higher relative to other reports (Table 3). The importance of _2_ and DO_2_ in our study is further supported by the positive linear regression between the main metabolic parameters and oxygen delivery and consumption. Indeed, early restoration of liver function and increased liver metabolism occurred in the BLOOD group with a better preservation of the energetic pool. The improved glucose and lactate uptake, and the increased glycogen and ATP tissue content, further indicate an upregulated metabolic status in the BLOOD group. The reduced potassium uptake and significant increase of W/D ratio at the end of perfusion (compared to SCS) in DMEM group likely depends on a more severe liver damage [37]. Furthermore, the direct relation between DO_2_/$\dot{VO}$_2_ and metabolic parameters suggests that the total amount of oxygen transported (DO_2_) results in an increased ability to use oxygen ($\dot{VO}$_2_) to generate energy (ATP) to sustain hepatocellular metabolism.

These data contradict the assumption of Mischinger and colleagues [31]. Furthermore, we deem that if DO_2_ is not optimized with an OxC, NMP, or NMRP could result in a suboptimal ex-situ reperfusion procedure. The reason for the disparity between our data and previous observations [32,38] could depend on differences in ex-situ perfusion or other procedural steps.

The importance of adequate DO_2_ is further highlighted in the rewarming phase. The BLOOD group showed an early $\dot{VO}$_2_ peak with early lactate normalization. A higher DO_2_ could help to overcome the oxygen debt accumulated during ischemia [17]. Although high O_2_ levels during reperfusion could potentially lead to oxidative tissue injury causing a more pronounced tissue damage [39], we did not observe any change in this direction.

The safety of human RBCs used in this model is demonstrated by the absence of endothelial damage, sinusoid obstruction, extravascular hemorrhage, or parenchymal damage. An adequate wash-out of rat blood during perfusion and the virtual absence of human plasma can avoid incompatibility reactions and, therefore, reduce species-specific antibody reactions.

Furthermore, in our study, histological data confirm the preserved graft viability indicated by the metabolic evaluation. As recently suggested [40], we used a combination of different biomarkers to assess graft quality, and some of them (glucose and potassium uptake ratio, citrate clearance, hyaluronic acid clearance) are described in this research for the first time. Their interaction could be used in clinical or preclinical settings to evaluate graft viability.

Some limitations of the study deserve comment. First of all, the duration of the NMP was relatively short if compared to a clinical setting. While accelerated clearance and sinusoidal trapping of human RBCs were observed in rat transfusion models [41], we did not observe significant hemolysis due to human blood use, and no red blood cells were trapped within sinusoidal spaces. However, the exploitation of these types of perfusate should be tested in prolonged perfusion to confirm absence of blood related liver damage. Non-cellular hemoglobin was recently used in clinical NMP [18], and it should be tested in this setting.

In conclusion, this study provides a detailed description of a small animal model of NMP characterized by optimized liver function, liver metabolism, and absence of injury. Results suggest that OxC may be adopted with NMP or NMRP, in particular with expanded criteria grafts, to avoid the detrimental impact of low DO_2_. In compliance with the 3R principle, human RBCs can be safely used to improve DO_2_, but additional studies are needed to confirm the safety, effectiveness, and optimal administration protocol.

## Figures and Tables

**Figure 1 jcm-08-01918-f001:**
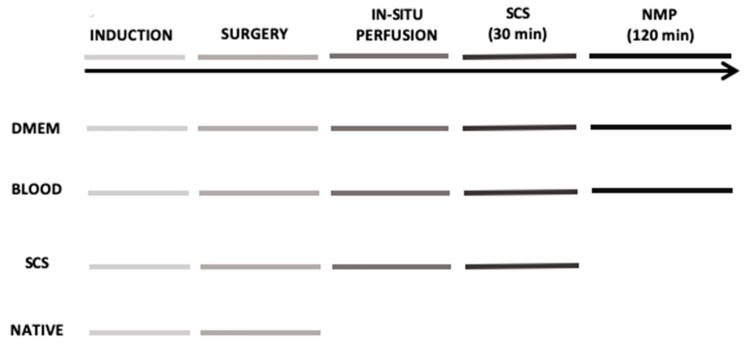
Schematic representation of the 4 experimental groups. DMEM and BLOOD groups performed all six steps of the timeline. Static cold storage (SCS) group underwent to all steps except for the normothermic machine perfusion, while Native group only to general anesthesia and surgery with liver tissue sampled at the end of the experiment.

**Figure 2 jcm-08-01918-f002:**
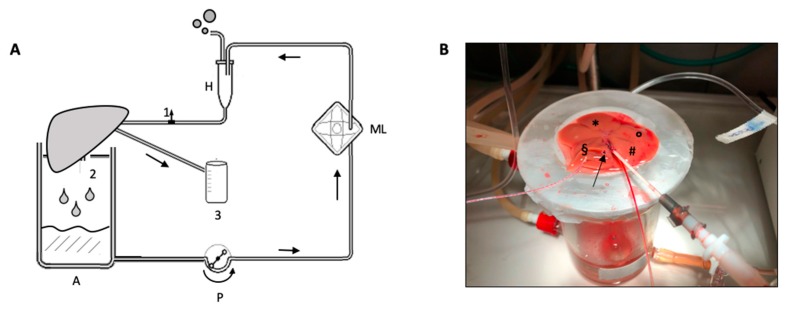
(**A**) Schematic representation of normothermic machine perfusion circuit. 1: pre-liver sampling stopcock; 2: post-liver sampling from IVC; 3: bile collection cuvette; A: heated glass reservoir; P: peristaltic pump; ML: membrane lung; H: heated exchanger and bubble trap. (**B**) Detail of the portal vein and bile duct cannula during ex-situ perfusion. Liver was laid on the diaphragmatic surface. Arrow: inferior vena cava; § right lateral lobe; * median lobe; ° left lateral lobe; ^#^ caudate lobe.

**Figure 3 jcm-08-01918-f003:**
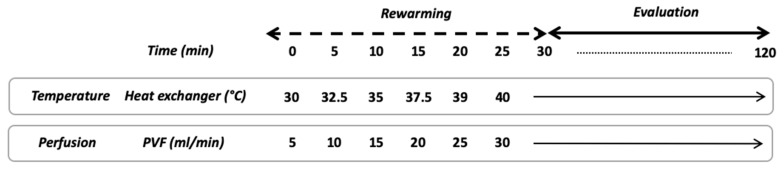
Schematic overview of NMP protocol. PVF, portal vein flow.

**Figure 4 jcm-08-01918-f004:**
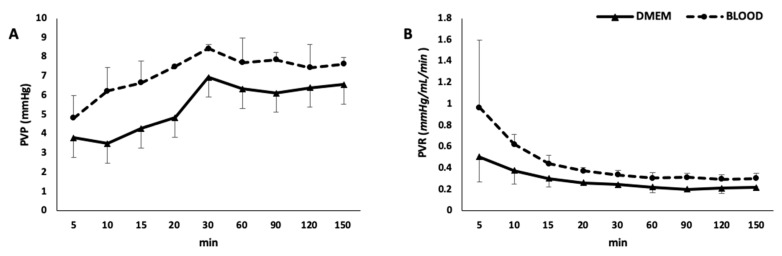
Portal vein pressure (**A**) and resistances (**B**)**.** PVP, portal vein pressure; PVR, portal vein resistances.

**Figure 5 jcm-08-01918-f005:**
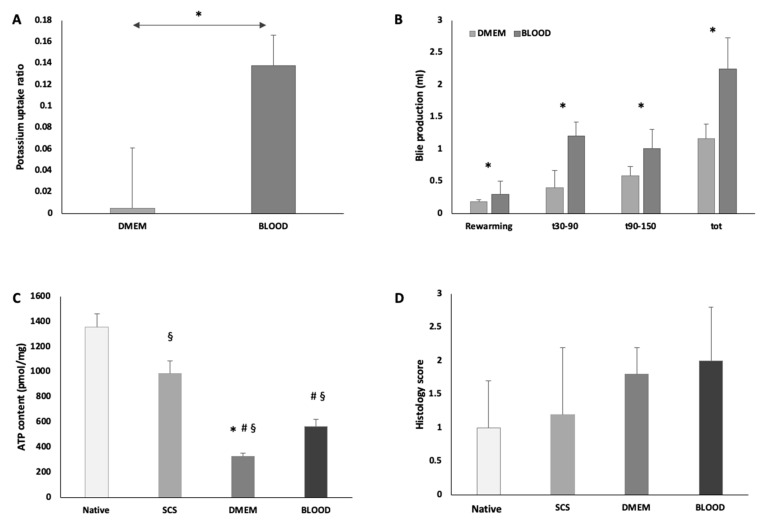
During ex-situ normothermic perfusion BLOOD group showed an increased potassium uptake ratio (* *p* < 0.05 DMEM vs. BLOOD) (**A**) and an increased bile production (* *p* < 0.05 DMEM vs. BLOOD) (**B**). These data suggest a more prompt and improved graft functional recovery, supported by the improved energetic pool at the end of NMP (*p* < 0.05, ^§^ vs. native, ^#^ vs. SCS, * vs. BLOOD) (**C**). NMP preserved parenchymal architecture in both experimental groups. In BLOOD group, the use of human red blood cells did not cause sinusoidal or parenchymal damage, consistently no trapped human red blood cells were found in sinusoids (**D**). Data are shown as mean ±SEM.

**Figure 6 jcm-08-01918-f006:**
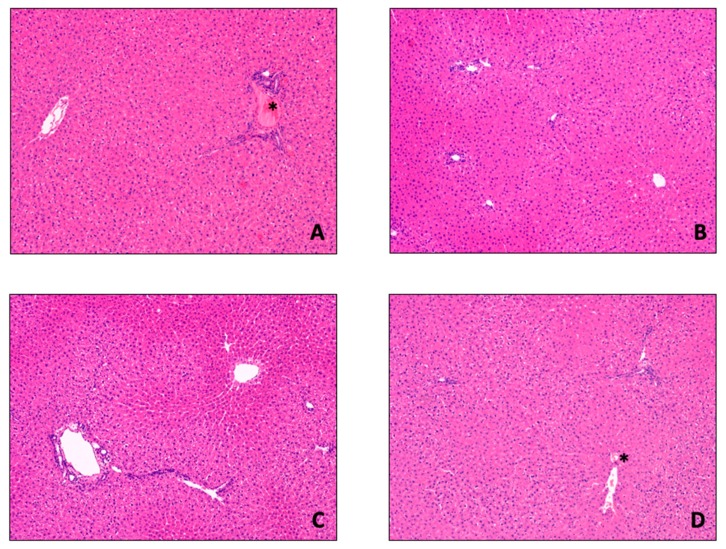
Histological examination of liver biopsies. Liver histological architecture was preserved in the four study groups ((**A**) Native (untreated/control) group; (**B**) SCS group; (**C**) DMEM group; (**D**) BLOOD group). As expected, red blood cells (*) can be found in Native and BLOOD group due to the presence of whole blood and blood based perfusate, respectively. Consistently, there were no hemorrhages or sinusoidal obstruction in BLOOD compared to the other groups. Sinusoidal architecture was preserved irrespective of the type of perfusion, as well as the rate of hepatocytes ploidy (6.2 ± 1.2/HPF). Hematoxylin and Eosin staining, 100× original magnification.

**Figure 7 jcm-08-01918-f007:**
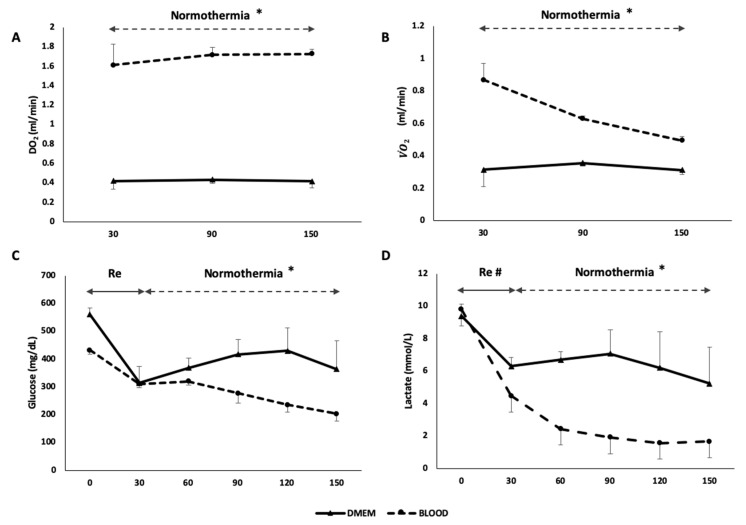
The use of human red blood cells as oxygen carrier resulted in an increased DO_2_ in BLOOD group (* *p* < 0.05) (**A**) and a consequent improved $\dot{VO}$_2_ (* *p* < 0.001) (**B**). Glucose was absorbed during normothermic phase in BLOOD group (* *p* < 0.001) (**C**) and lactates metabolism was increased during the whole procedure in BLOOD group (^#^
*p* < 0.05) (**D**). These parameters suggest an improved metabolic function of the graft when a higher oxygen content was provided during perfusion. Re, rewarming; DO_2_, delivery of oxygen; $\dot{VO}$_2_, oxygen consumption. Data are shown as mean ± SEM.

**Figure 8 jcm-08-01918-f008:**
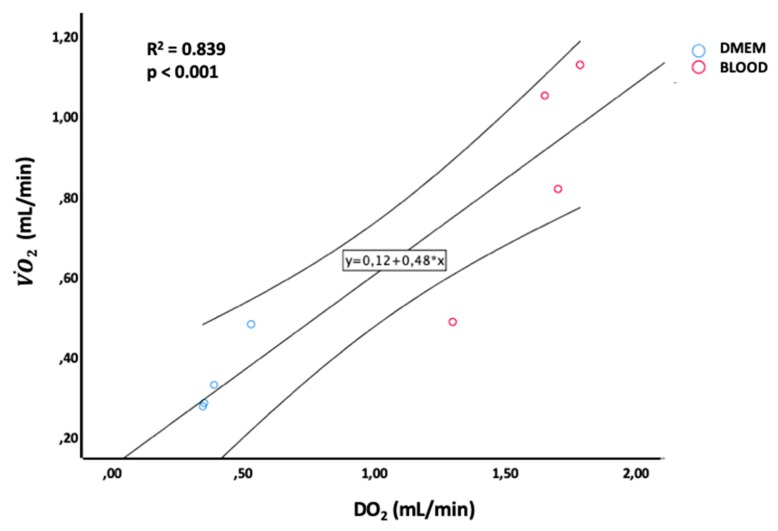
Linear regression between DO_2_ and $\dot{VO}$_2_ levels during ex-situ normothermic perfusion.

**Table 1 jcm-08-01918-t001:** Characteristics of the perfusion fluid in the two experimental groups: baseline blood-gas analyses before liver graft connection to NMP. Hb, hemoglobin; Htc, hematocrit; HbO_2_, hemoglobin saturation; Gluc, glucose; Lac, lactate; na, not applicable. Results are expressed as mean ± SEM.

	DMEM (*n* = 5)	BLOOD (*n* = 4)	*p*
**pH**	7.38 ± 0.09	7.18 ± 0.09	0.042
**pCO_2_, mmHg**	40.5 ± 8	49.5 ± 13	0.231
**pO_2_, mmHg**	540.25 ± 26	446 ± 179	0.317
**Hb, g/dL**	-	4.4 ± 0.57	na
**Htc, %**	-	14 ± 2.00	na
**HbO_2_, %**	0	96.25 ± 0.07	na
**Gluc, g/dL**	80 ± 0.50	136 ± 53	0.026
**Lac, mmol/L**	-	2.3 ± 1.7	na
**K^+^, mmol/L**	4.15 ± 0.05	4.5 ± 1	0.427
**Na^+^, mmol/L**	149 ± 1	147 ± 3	0.553
**Ca^2+^, mmol/L**	0.72 ± 0.28	0.70 ± 0.05	0.342
**Cl^−^, mmol/L**	121 ± 2.38	117 ± 5.00	0.522
**HCO_3_^−^, mmol/L**	23 ± 1.87	15 ± 1.00	0.011

**Table 2 jcm-08-01918-t002:** Main metabolic and biological characteristics of normothermic machine perfusion during normothermic phase in the two study groups. * compared to SCS group (2.872 ± 0.128) *p* = 0.029 in DMEM and *p* = 0.275 in BLOOD. ^ compared to SCS group (985.9 ± 102.6 pmol/mg) *p* = 0.001 in DMEM and *p* = 0.012 in BLOOD. Results are expressed as mean ± SEM.

Normothermic Phase (30-150 min)					
	Min	DMEM (n = 5)	BLOOD (n = 4)	*p* (Graft)	*p* (Time)
**AST, U/L/g**	30	0.862 ± 0.769	0.739 ± 0.215	0.358	
	150	2.130 ± 1.591	2.273 ± 1.47		
**Potassium mEq/L**	30	5.45 ± 0.17	5.10 ± 0.62	0.657	0.01
	150	5.40 ± 0.43	4.40 ± 0.61		
**Potassium uptake ratio**		0.006 ± 0.051	0.138 ± 0.028	0.03	na
**Bile, g**	tot	1.165 ± 0.22	2.25 ± 0.48	0.237	
**W/D ratio ***		3.117 ± 0.136	2.995 ±. 0.85	0.208	
**ATP, pmol/mg ^**		327.3 ± 26.6	565.8 ± 56.6	0.031	
**Glucose, mg/dL**	30	313 ± 59	310 ± 12	0.008	<0.001
	150	364 ± 101	202 ± 26		
**Glucose uptake ratio**		−0.136 ± 0.154	0.346 ± 0.079	0.029	
**Lactate, mmol/L**	30	6.3 ± 0.5	4.5 ± 1.0	0.002	<0.001
	150	5.24 ± 1.2	1.6 ± 0.6		
**Lactate uptake ratio**		0.062 ± 0.359	0.643 ± 0.27	0.022	na
**Citrate**	30	na	0.37 ± 0.05		0.034
	150	na	0.29 ± 0.05		
**Citrate uptake**		na	0.228 ± 0.127		

**Table 3 jcm-08-01918-t003:** Main metabolic parameters reported in literature during ex-situ normothermic perfusion on rats. Htc, hematocrit; pO_2_, partial pressure of oxygen in perfusate; $\dot{VO}$_2_, oxygen consumption; RBC, packed red blood cells or centrifugated red blood cells.

Author	Year	Blood Type	Htc	Blood Origin	pO_2_	VO˙2 (max)(mL/min)
Dutkowski P	2006	full blood	6.4	rat	450	0.26
Izamis ML	2013	RBC	18	rat		0.5
Schlegel A	2014	full blood	15	rat	300–375	
Tolboom H	2012	RBC	18	rat		0.5
Tolboom H	2008	RBC	18	rat		0.4
Westerkamp AC	2015	RBC	25	human	450–600

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
