# Peer review of "Human Red Blood Cells as Oxygen Carriers to Improve Ex-Situ Liver Perfusion in a Rat Model"

_jcm, 2019, doi:10.3390/jcm8111918_

Round 1

Reviewer 1 Report

Article Review

„Human red blood cells as oxygen carriers to improve 3 ex-situ liver perfusion in a rat model”

The article entilted: „Human red blood cells as oxygen carriers to improve 3 ex-situ liver perfusion in a rat model” written by team of Dondossola Daniele is extremely interesting and the subject of the research is based on the field of hepatology. The subject seems to be very important since transplantology is one of the fastest growing area of medicine and what is more, more and more patients suffer from various diseases requiring transplantation, inclusing liver transplantation.

Therefore, as far as I am concerned, I reccomend to publish the current article, however, the manuscript needs some minor revision.

Abstract should be written in more understandable style. The experimental groups should be described more clearly, 1) native ..of what??; 2) static cold storage ..of what?? This should be improved to make the manuscript more understandable, since the manuscript includes so many results and so many aspects and it is a little bit complicated.

Material and methods: I suggest to change  the group order according to the description in the text in the Figure 1.

Results are described properly. Are there any histological pictures available? I suggest to add them to the article to show the comparison of livers from each group.

Discussion is performed properly and it ends with the proper conclusion.

At the end, as far as I am concerned, I would use coursive when using latine words as” ex vivo, in situ and so on” , including the title.

Author Response

We thank the reviewer for the helpful comments. Our responses to the points raised and a description of the changes made accordingly are shown below. Comments are typed in bold and responses in regular typeface.

Therefore, as far as I am concerned, I recommend to publish the current article, however, the manuscript needs some minor revision.  

Abstract should be written in more understandable style. The experimental groups should be described more clearly, 1) native ..of what??; 2) static cold storage ..of what?? This should be improved to make the manuscript more understandable, since the manuscript includes so many results and so many aspects and it is a little bit complicated.

We reviewed the abstract to make it more clear. Native was changed in “native (untreated/control)” (line 36), while static cold storage was changed in “liver static cold storage (SCS) 30 min without NMP” (line 37).

Material and methods: I suggest to change the group order according to the description in the text in the Figure 1.

We reviewed the group order as requested (line 103-109).

Results are described properly. Are there any histological pictures available? I suggest to add them to the article to show the comparison of livers from each group.

Histological pictures were added as figure (figure 6; line 437), thank you for the suggestion.

At the end, as far as I am concerned, I would use coursive when using latine words as” ex vivo, in situ and so on” , including the title.

Changes were made according to the requested style.

Reviewer 2 Report

The research article “Human red blood cells as oxygen carriers to improve 2 ex-situ liver perfusion in a rat model”, by Dondossola and colleagues documents the utility of using human erythrocites to optimize the ex-situ liver perfusion in a rat model. The experimental design is well organized and methodologically correct. It is also respectful of the international guidelines for in vivo research. Very few issues concern

(a) the histological samples: hepatocytes may increase ploidy (diploid, tetraploid, and even octaploid) depending on the age, paraphysiological status and experimental conditions. Do authors observe differences in the hepatocytes over the different 4 groups? 

(b) Could be auspicable that this model is complemented with and further validated by imaging methods.

Minor concerns:

Abstract:

please specify “V02” stands for

Materials and methods:

p 2 line 85: better use suppression than sacrifice

p 4 lines 144-154: are in italic, please change into regular

Fig. 2B: to help the reader, please indicate the view (dorsal view of liver?) and the lobes (by using arrows)

Discussion:

p13 line 400: please delete "on a model of...perfusion", it is quite redundant

Author Response

We thank the reviewer for the helpful comments. Our responses to the points raised and a description of the changes made accordingly are shown below. Comments are typed in bold and responses in regular typeface.

The research article “Human red blood cells as oxygen carriers to improve 2 ex-situ liver perfusion in a rat model”, by Dondossola and colleagues documents the utility of using human erythrocites to optimize the ex-situ liver perfusion in a rat model. The experimental design is well organized and methodologically correct. It is also respectful of the international guidelines for in vivo research. Very few issues concern.

The histological samples: hepatocytes may increase ploidy (diploid, tetraploid, and even octaploid) depending on the age, paraphysiological status and experimental conditions. Do authors observe differences in the hepatocytes over the different 4 groups?

We reviewed the histological samples to identify differences in ploidy. Two different pathologist found only diploid hepatocyte (mean 6.25/HPF), without differences in the four study groups. We added this information in the caption of figure 6 (line 444-445)

Could be auspicable that this model is complemented with and further validated by imaging methods.

We added histological pictures to offer a deeper explanation of our results. Unfortunately other imaging techniques (such as MRI or CT) are not available in our preclinical facility, thus it is not possible to add these information to this article. Furthermore, we believe that imaging methods cannot add further evidences to this model. 

Abstract: please specify “V02” stands for.

We reviewed this abbreviation (line 40-41)

Materials and methods: p 2 line 85: better use suppression than sacrifice

We change sacrifice with suppression (line 109)

p 4 lines 144-154: are in italic, please change into regular

Italic was changed in regular, thank you for the suggestion

2B: to help the reader, please indicate the view (dorsal view of liver?) and the lobes (by using arrows)

We have specified the view of the liver and we named the lobes (line 258-260).

Discussion: p13 line 400: please delete "on a model of...perfusion", it is quite redundant

We reviewed the discussions (line 507).
